# Aligning Visual Regions and Textual Concepts for Semantic-Grounded Image Representations

**Fenglin Liu[1]**[*] **Yuanxin Liu[3,4]**[*] **Xuancheng Ren[2]**[*] **Xiaodong He[5], Xu Sun[2]**
[1]ADSPLAB, School of ECE, Peking University, Shenzhen, China
[2]MOE Key Laboratory of Computational Linguistics, School of EECS, Peking University
[3]Institute of Information Engineering, Chinese Academy of Sciences
[4]School of Cyber Security, University of Chinese Academy of Sciences
[5]JD AI Research
{fenglinliu98, renxc, xusun}@pku.edu.cn, liuyuanxin@iie.ac.cn
xiaodong.he@jd.com

## Abstract

In vision-and-language grounding problems, fine-grained representations of the image are considered to be of paramount importance. Most of the current systems incorporate visual features and textual concepts as a sketch of an image. However, plainly inferred representations are usually undesirable in that they are composed of separate components, the relations of which are elusive. In this work, we aim at representing an image with a set of integrated visual regions and corresponding textual concepts, reflecting certain semantics. To this end, we build the Mutual Iterative Attention (MIA) module, which integrates correlated visual features and textual concepts, respectively, by aligning the two modalities. We evaluate the proposed approach on two representative vision-and-language grounding tasks, i.e., image captioning and visual question answering. In both tasks, the semantic-grounded image representations consistently boost the performance of the baseline models under all metrics across the board. The results demonstrate that our approach is effective and generalizes well to a wide range of models for image-related applications.[2]

## 1 Introduction

Recently, there is a surge of research interest in multidisciplinary tasks such as image captioning [7] and visual question answering (VQA) [3], trying to explain the interaction between vision and language. In image captioning, an intelligence system takes an image as input and generates a description in the form of natural language. VQA is a more challenging problem that takes an extra question into account and requires the model to give an answer depending on both the image and the question. Despite their different application scenarios, a shared goal is to understand the image, which necessitates the acquisition of grounded image representations.

In the literature, an image is typically represented in two fundamental forms: visual features and textual concepts (see Figure 1). **Visual Features** [30, 2, 18] represent an image in the vision domain and contain abundant visual information. For CNN-based visual features, an image is split into equally-sized visual regions without encoding global relationships such as position and adjacency. To obtain better image representations with respect to concrete objects, RCNN-based visual features that are defined by bounding boxes of interests are proposed. Nevertheless, the visual features are based

---

[*]Equal contribution.
[2]The code is available at https://github.com/fenglinliu98/MIA

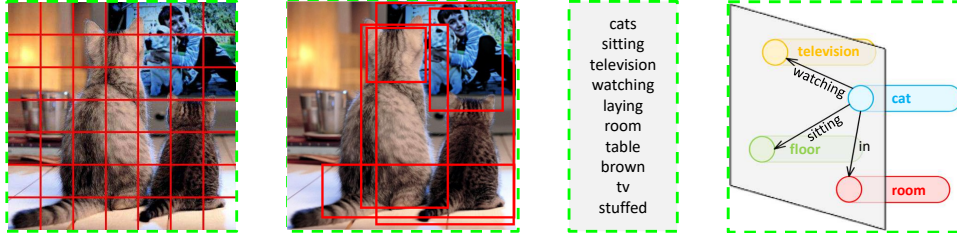

Figure 1: Illustrations of commonly-used image representations (from left to right): CNN-based grid visual features, RCNN-based region visual features, textual concepts, and scene-graphs.

on regions and are not associated with the actual words, which means the semantic inconsistency between the two domains has to be resolved by the downstream systems themselves. **Textual Concepts** [8, 35, 31] represent an image in the language domain and introduce semantic information. They consist of unordered visual words, irrespective of affiliation and positional relations, making it difficult for the system to infer the underlying semantic and spatial relationships. Moreover, due to the lack of visual reference, some concepts may induce semantic ambiguity, e.g., the word *mouse* can either refer to a mammal or an electronic device. **Scene-Graphs** [34] are the combination of the two kinds of representations. They use region-based visual features to represent the objects and textual concepts to represent the relationships. However, to construct a scene-graph, a complicated pipeline is required and error propagation cannot be avoided.

For image representations used for text-oriented purposes, it is often desirable to integrate the two forms of image information. Existing downstream systems achieve that by using both kinds of image representations *in the decoding process*, mostly ignoring the innate alignment between the modalities. As the semantics of the visual features and the textual concepts are usually inconsistent, the systems have to devote themselves to learn such alignment. Besides, these representations only contain local features, lacking global structural information. Those problems make it hard for the systems to understand the image efficiently.

In this paper, we work toward constructing integrated image representations from vision and language *in the encoding process*. The objective is achieved by the proposed Mutual Iterative Attention (MIA) module, which aligns the visual features and textual concepts with their relevant counterparts in each domain. The motivation comes from the fact that correlated features in one domain can be linked up by a feature in another domain, which has connections with all of them. In implementation, we perform mutual attention iteratively between the two domains to realize the procedure without annotated alignment data. The visual receptive fields gradually concentrate on salient visual regions, and the original word-level concepts are gradually merged to recapitulate corresponding visual regions. In addition, the aligned visual features and textual concepts provide a more clear definition of the image aspects they represent.

The contributions of this paper are as follows:

- For vision-and-language grounding problems, we introduce integrated image representations based on the alignment between visual regions and textual concepts to describe the salient combination of local features in a certain modality.

- We propose a novel attention-based strategy, namely the Mutual Iterative Attention (MIA), which uses the features from the other domain as the guide for integrating the features in the current domain without mixing in the heterogeneous information.

- According to the extensive experiments on the MSCOCO image captioning dataset and VQA v2.0 dataset, when equipped with the MIA, improvements on the baselines are witnessed in all metrics. This demonstrates that the semantic-grounded image representations are effective and can generalize to a wide range of models.

## 2    Approach

The proposed approach acts on plainly extracted image features from vision and language, e.g., convolutional feature maps, regions of interest (RoI), and visual words (textual concepts), and refines

those features so that they can describe visual semantics, i.e., meaningful compositions of such features, which are then used in the downstream tasks to replace the original features. Figure 2 gives an overview and an example of our approach.

## 2.1 Visual Features and Textual Concepts

Visual features and textual concepts are widely used [35, 33, 15, 21] as the information sources for image-grounded text generation. In common practice, visual features are extracted by ResNet [10], GoogLeNet [27] and VGG [25], and are rich in low-level visual information [31]. Recently, more and more work adopted regions of interest (RoI) proposed by RCNN-like models as visual features, and each RoI is supposed to contain a specific object in the image. Textual concepts are introduced to compensate the lack of high-level semantic information in visual features [8, 31, 35]. Specifically, they consist of visual words that can be objects (e.g., *dog, bike*), attributes (e.g., *young, black*) and relations (e.g., *sitting, holding*). The embedding vectors of these visual words are then taken as the textual concepts. It is worth noticing that to obtain visual features and textual concepts, only the image itself is needed as input, and no external text information about the image is required, meaning that they can be used for any vision-and-language grounding problems. In the following, we denote the visual features and textual concepts for an image as $I$ and $T$, respectively.

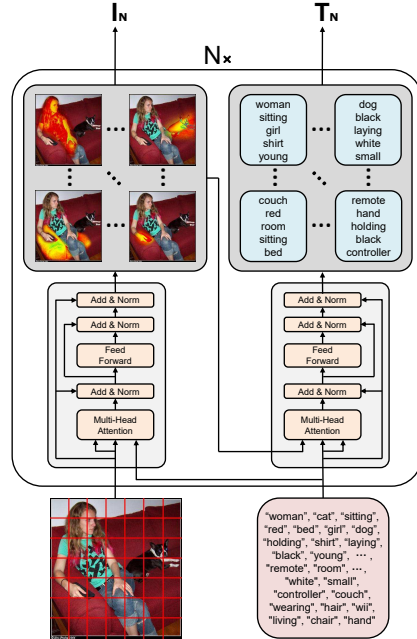

Figure 2: Overview of our approach. We take as input visual features and textual concepts (the lower) and repeat a mutual attention mechanism (the middle) to combine the local features from each domain, resulting in integrated image representations reflecting certain semantics of the image (the upper).

## 2.2 Learning Alignment

To form the alignment between the visual regions and the textual words, we adopt the attention mechanism from Vaswani et al. [28], which is designed initially to obtain contextual representations for sentences in machine translation and has proven to be effective in capturing alignment of different languages and structure of sentences.

### 2.2.1 Mutual Attention

Mutual Attention contains two sub-layers. The first sub-layer makes use of multi-head attention to learn the correlated features in a certain domain by querying the other domain. The second sub-layer uses feed-forward layer to add sufficient expressive power.

The multi-head attention is composed of $k$ parallel heads. Each head is formulated as a scaled dot-product attention:

$$\text{Att}_i(Q, S) = \text{softmax}\left(\frac{QW_i^Q(SW_i^K)^\mathsf{T}}{\sqrt{d_k}}\right)SW_i^V, \quad i = 1, \ldots, k \tag{1}$$

where $Q \in \mathbb{R}^{m \times d_h}$ and $S \in \mathbb{R}^{n \times d_h}$ stand for $m$ querying features and $n$ source features, respectively; $W_i^Q, W_i^K, W_i^V \in \mathbb{R}^{d_h \times d_k}$ are learnable parameters of linear transformations; $d_h$ is the size of the input features and $d_k = d_h/k$ is the size of the output features for each attention head. Results from each head are concatenated and passed through a linear transformation to construct the output:

$$\text{MultiHeadAtt}(Q, S) = [\text{Att}_1(Q, S), \ldots, \text{Att}_k(Q, S)]W^O \tag{2}$$

where $W^O \in \mathbb{R}^{d_h \times d_h}$ is the parameter to be learned. The multi-head attention integrates $n$ source features into $m$ output features in the order of querying features. To simplify computation, we keep $m$ the same as $n$.

Following the multi-head attention is a fully-connected network, defined as:

$$\text{FCN}(X) = \max\left(0, XW^{(1)} + b^{(1)}\right)W^{(2)} + b^{(2)} \tag{3}$$

where $W^{(1)}$ and $W^{(2)}$ are matrices for linear transformation; $b^{(1)}$ and $b^{(2)}$ are the bias terms. Each sub-layer is followed by an operation sequence of dropout [26], shortcut connection[3] [10], and layer normalization [4].

Finally, the mutual attention is conducted as:

$$I' = \text{FCN}(\text{MultiHeadAtt}(T, I)), \quad T' = \text{FCN}(\text{MultiHeadAtt}(I', T)) \tag{4}$$

i.e., visual features are first integrated according to textual concepts, and then textual concepts are integrated according to integrated visual features. It is worth noticing that it is also possible to reverse the order by first constructing correlated textual concepts. However, in our preliminary experiments, we found that the presented order performs better. The related results and explanations are given in the supplementary materials for reference.

The knowledge from either domain can serve as the guide for combining local features and extracting structural relationships of the other domain. For example, as shown by the upper left instance of the four instances in Figure 2, the textual concept *woman* integrates the regions that include the woman, which then draw in textual concepts *sitting, girl, shirt, young*. In addition, the mutual attention aligns the two kinds of features, because for the same position in the two feature matrices, the integrated visual feature and the integrated textual concept are co-referential and represent the same high-level visual semantics. This approach also ensures that the refined visual features only contain homogeneous information because the information from the other domain only serves as the attentive weight and is not part of the final values.

### 2.2.2 Mutual Iterative Attention

To refine both the visual features and the textual concepts, we propose to perform mutual attention iteratively. The process in Eq. (4) that uses the original features is considered as the first round:

$$I_1 = \text{FCN}(\text{MultiHeadAtt}(T_0, I_0)), \quad T_1 = \text{FCN}(\text{MultiHeadAtt}(I_1, T_0)) \tag{5}$$

where $I_0, T_0, I_1$ and $T_1$ represent the original visual features, the original textual concepts, the macro visual features, and the macro textual concepts, respectively. By repeating the same process for $N$ times, we obtain the final outputs of the two stacks:

$$I_N = \text{FCN}(\text{MultiHeadAtt}(T_{N-1}, I_{N-1})), \quad T_N = \text{FCN}(\text{MultiHeadAtt}(I_N, T_{N-1})) \tag{6}$$

It is important to note that in each iteration, the parameters of the mutual attention are shared. However, as in each iteration more information is integrated into each feature, it is possible that iterating too many times would cause *the over-smoothing problem* that all features represent essentially the same and the overall semantics of the image. To avoid such problem, we apply the aforementioned post-processing operations to the output of each layer, but with the shortcut connection from the input of each layer (not the sub-layer). The shortcut serves as a semantic anchor that prevents the peripheral information from extending the pivotal visual or textual features too much and keeps the position of each semantic-grounded feature stable in the feature matrices.

For the downstream tasks consuming both visual features and textual concepts of images, $I_N$ and $T_N$ can be directly used to replace the original features, respectively, because the number and the size of the features are kept through the procedure. However, since the visual features and the textual concepts are already aligned, we can directly add them up to get the output that makes the best of their respective advantages, even for the tasks that originally only consumes one kind of image representations:

$$\text{MIA}(I, T) = \text{LayerNorm}(I_N + T_N) \tag{7}$$

As a result, the refined features overcome the aforementioned weaknesses of existing image representations, providing a better start point for downstream tasks. For tasks using both kinds features, each kind feature can be replaced with MIA-refined features.

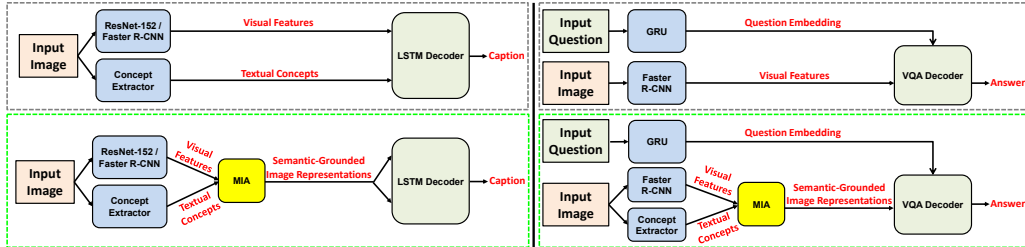

Figure 3: Illustration of how to equip the baseline models with our MIA. MIA aligns and integrates the original image representations from two modalities. Left: For image captioning, the semantic-grounded image representations are used to replace both kinds of original image features. Right: For VQA, MIA only substitutes the image representations, and the question representations are preserved.

As annotated alignment data is not easy to obtain and the alignment learning lacks direct supervision, we adopt the distantly-supervised learning and refine the integrated image representations with downstream tasks. As shown by previous work [28], when trained on machine translation, the attention can learn correlation of words quite well. As the proposed method focuses on building semantic-grounded image representations, it can be easily incorporated in the downstream models to substitute the original image representations, which in turn provides supervision for the mutual iterative attention. Specifically, we experiment with the task of image captioning and VQA. To use the proposed approach, MIA is added to the downstream models as a preprocessing component. Figure 3 illustrates how to equip the baseline systems with MIA, through two examples for image captioning and VQA, respectively. As we can see, MIA substitutes the original image representations with semantic-grounded image representations. For VQA, the question representations are preserved. Besides, MIA does not affect the original experimental settings and training strategies.

## 3 Experiment

We evaluate the proposed approach on two multi-modal tasks, i.e., image captioning and visual question answering (VQA). We first conduct experiments on representative systems that use different kinds of image representations to demonstrate the effectiveness of the proposed semantic-grounded image representations, and then provide analysis of the key components of the MIA module.

Before introducing the results and the analysis, we first describe some common settings. The proposed MIA relies on both visual features and textual concepts to produce semantic-grounded image representations. Considering the diverse forms of the original image representations, unless otherwise specified, they are obtained as follows: (1) the grid visual features are from a ResNet-152 pretrained on ImageNet, (2) the region-based visual features are from a variant of Faster R-CNN [23], which is provided by Anderson et al. [2] and pre-trained on Visual Genome [13], and (3) the textual concepts are extracted by a concept extractor in Fang et al. [8] trained on the MSCOCO captioning dataset using Multiple Instance Learning [36]. The number of textual concepts is kept the same as the visual features, i.e., 49 for grid visual features and 36 for region visual features, by keeping only the top concepts. The settings of MIA are the same for the two tasks, which reflects the generality of our method. Particularly, we use 8 heads ($k = 8$) and iterate twice ($N = 2$), according to the performance on the validation set. For detailed settings, please refer to the supplementary material.

### 3.1 Image Captioning

**Dataset and Evaluation Metrics.** We conduct experiments on the MSCOCO image captioning dataset [7] and use SPICE [1], CIDEr [29], BLEU [22], METEOR [5] and ROUGE [14] as evaluation metrics, which are calculated by MSCOCO captioning evaluation toolkit [7]. Please note that following common practice [17, 2, 16], we adopt the dataset split from Karpathy and Li [11] and the results are not comparable to those from the online MSCOCO evaluation server.

**Baselines.** Given an image, the image captioning task aims to generate a descriptive sentence accordingly. To evaluate how the proposed semantic-grounded image representation helps the downstream tasks, we first design five representative baseline models that take as input different image representations based on previous work. They are (1) Visual Attention, which uses grid visual

Table 1: Results of the representative systems on the image captioning task.

| Methods | BLEU-1 | BLEU-2 | BLEU-3 | BLEU-4 | METEOR | ROUGE | CIDEr | SPICE |
|---|---|---|---|---|---|---|---|---|
| Visual Attention | 72.6 | 56.0 | 42.2 | 31.7 | 26.5 | 54.6 | 103.0 | 19.3 |
| w/ MIA | **74.5** | **58.4** | **44.4** | **33.6** | **26.8** | **55.8** | **106.7** | **20.1** |
| Concept Attention | 72.6 | 55.9 | 42.5 | 32.5 | 26.5 | 54.4 | 103.2 | 19.4 |
| w/ MIA | **73.8** | **57.4** | **43.8** | **33.6** | **27.1** | **55.3** | **107.9** | **20.3** |
| Visual Condition | 73.3 | 56.9 | 43.4 | 33.0 | 26.8 | 54.8 | 105.2 | 19.5 |
| w/ MIA | **73.9** | **57.3** | **43.9** | **33.7** | **26.9** | **55.1** | **107.2** | **19.8** |
| Concept Condition | 72.9 | 56.2 | 42.8 | 32.7 | 26.4 | 54.4 | 104.4 | 19.3 |
| w/ MIA | **73.9** | **57.3** | **43.9** | **33.7** | **26.9** | **55.1** | **107.2** | **19.8** |
| Visual Regional Attention | 75.2 | 58.9 | 45.2 | 34.7 | 27.6 | 56.0 | 111.2 | 20.6 |
| w/ MIA | **75.6** | **59.4** | **45.7** | **35.4** | **28.0** | **56.4** | **114.1** | **21.1** |

Table 2: Evaluation of systems that use reinforcement learning on the MSCOCO image captioning dataset.

| Methods | BLEU-4 | METEOR | ROUGE | CIDEr | SPICE |
|---|---|---|---|---|---|
| Up-Down | 36.5 | 28.0 | 57.0 | 120.9 | 21.5 |
| w/ MIA | **37.0** | **28.2** | **57.4** | **122.2** | **21.7** |
| Transformer | 39.0 | 28.4 | 58.6 | 126.3 | 21.7 |
| w/ MIA | **39.5** | **29.0** | **58.7** | **129.6** | **22.7** |

Table 3: The overall accuracy on the VQA v2.0 test dataset.

| Methods | Test-dev | Test-std |
|---|---|---|
| Up-Down | 67.3 | 67.5 |
| w/ MIA | **68.8** | **69.1** |
| BAN | 69.6 | 69.8 |
| w/ MIA | **70.2** | **70.3** |

features as the attention source for each decoding step, (2) Concept Attention, which uses textual concepts as the attention source, (3) Visual Condition, which takes textual concepts as extra input at the first decoding step but grid visual features in the following decoding steps, (4) Concept Condition, which, in contrast to Visual Condition, takes grid visual features at the first decoding step but textual concepts in the following decoding steps, and (5) Visual Regional Attention, which uses region-based visual features as the attention source. For those models, the traditional cross-entropy based training objective is used. We also check on the effect of MIA on more advanced captioning models, including (6) Up-Down [2], which uses region-based visual features, and (7) Transformer, which adapts the Transformer-Base model in Vaswani et al. [28] by taking the region-based visual features as input. Those advanced models adopt CIDEr-based training objective using reinforcement training [24].

**Results.** In Table 1, we can see that the models enjoy an increase of 2%~5% in terms of both SPICE and CIDEr, with the proposed MIA. Especially, "Visual Attention w/ MIA" and "Concept Attention w/ MIA" are able to pay attention to integrated representation collections instead of the separate grid visual features or textual concepts. Besides, the baselines also enjoy the benefit from the semantic-grounded image representations, which can be verified by the improvement of "Visual Regional Attention w/ MIA". The results demonstrate the effectiveness and universality of MIA. As shown in Table 2, the proposed method can still bring improvements to the strong baselines under the reinforcement learning settings. Besides, it also suggests that our approach is compatible with both the RNN based (Up-Down) and self-attention based (Transformer) language generators. We also investigate the effect of incorporating MIA with the scene-graph based model [32], the results are provided in the supplementary material, where we can also see consistent improvements. In all, the baselines are promoted in all metrics across the board, which indicates that the refined image representations are less prone to the variations of model structures (e.g., with or without attention, and the architecture of downstream language generator), hyper-parameters (e.g., learning rate and batch size), original image representations (e.g., CNN, RCNN-based visual features, textual concepts and scene-graphs), and learning paradigm (e.g., cross-entropy and CIDEr based objective).

## 3.2 Visual Question Answering

**Dataset and Evaluation Metrics.**

We experiment on the VQA v2.0 dataset [9], which is comprised of image-based question-answer pairs labeled by human annotators. The questions are categorized into three types, namely Yes/No,

Table 4: Ablation analysis of the proposed approach. As we can see, incorporating MIA-refined image representation from a single modality can also lead to overall improvements.

| Methods | BLEU-1 | BLEU-2 | BLEU-3 | BLEU-4 | METEOR | ROUGE | CIDEr | SPICE |
|---|---|---|---|---|---|---|---|---|
| Visual Attention | 72.6 | 56.0 | 42.2 | 31.7 | 26.5 | 54.6 | 103.0 | 19.3 |
| w/ $I_N$ | **74.7** | **58.5** | **44.6** | **33.7** | 26.5 | 55.2 | 105.7 | 19.6 |
| w/ MIA | 74.5 | 58.4 | 44.4 | 33.6 | **26.8** | **55.8** | **106.7** | **20.1** |
| Concept Attention | 72.6 | 55.9 | 42.5 | 32.5 | 26.5 | 54.4 | 103.2 | 19.4 |
| w/ $T_N$ | 73.7 | 57.0 | 43.4 | 33.1 | 26.8 | 55.0 | 106.5 | 20.0 |
| w/ MIA | **73.8** | **57.4** | **43.8** | **33.6** | **27.1** | **55.3** | **107.9** | **20.3** |

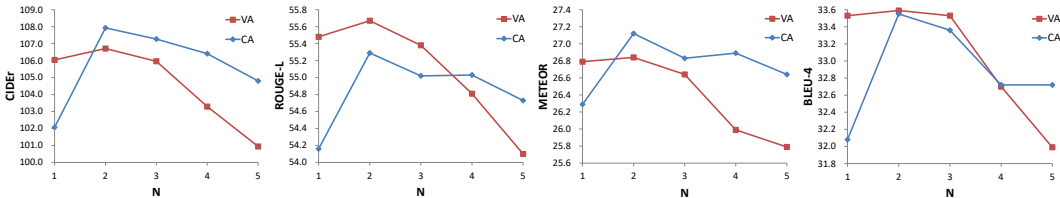

Figure 4: Model performance variation under different metrics with the increase of iteration times. VA and CA stand for Visual Attention and Concept Attention, respectively.

Number and other categories. We report the model performance based on overall accuracy on both the test-dev and test-std sets, which is calculated by the standard VQA metric [3].

**Baselines.** Given an image and a question about the image, the visual question answering task aims to generate the correct answer, which is modeled as a classification task. We choose Up-Down [2] and BAN [12] for comparison. They both use region-based visual features as image representations and GRU-encoded hidden states as question representations, and make classification based on their combination. However, Up-Down only uses the final sentence vector to obtain the weight of each visual region, while BAN uses a bilinear attention to obtain the weight for each pair of visual region and question word. BAN is the previous state-of-the-art on the VQA v2.0 dataset.

**Results.** As shown in Table 3, an overall improvement is achieved when applying MIA to the baselines, which validates that our method generalizes well to different tasks. Especially, on the answer type *Number*, the MIA promotes the accuracy of Up-Down from 47.5% to 51.2% and BAN from 50.9% to 53.1%. The significant improvements suggest that the refined image representations are more accurate in counting thanks to integrating semantically related objects.

### 3.3 Analysis

In this section, we analyze the effect of the proposed approach and provide insights of the MIA module, in an attempt to answer the following questions: (1) Is the mutual attention necessary for integrating semantically-related features? (2) Is the improvement spurious because MIA uses two kinds input features while some of the baseline models only use one? (3) How does the iteration time affect the alignment process? and (4) Does the mutual attention actually align the two modality?

**Effect of mutual attention.** Mutual attention serves as a way to integrate correlated features by aligning modalities, which is our main proposal. Another way to integrate features is to only rely on information from one domain, which can be achieved by replacing mutual attention with self-attention. However, this method is found to be less effective than MIA, scoring 96.6 and 105.4 for Visual Attention and Concept Attention, respectively, in terms of CIDEr. Especially, the performance of the Visual Attention has even been impaired, which suggests that only using information from one domain is insufficient to construct meaningful region or concept groups that are beneficial to describing images and confirms our main motivation. Besides, as the self-attention and the mutual attention shares the same multi-head attention structure, it also indicates that the improvement comes from the alignment of the two modalities rather than the application of the attention structure.

**Ablation Study.** As the deployment of MIA inevitably introduces information from the other modality, we conduct ablation studies to investigate whether the improvement is derived from the well-aligned and integrated image representations or the additional source information. As shown in

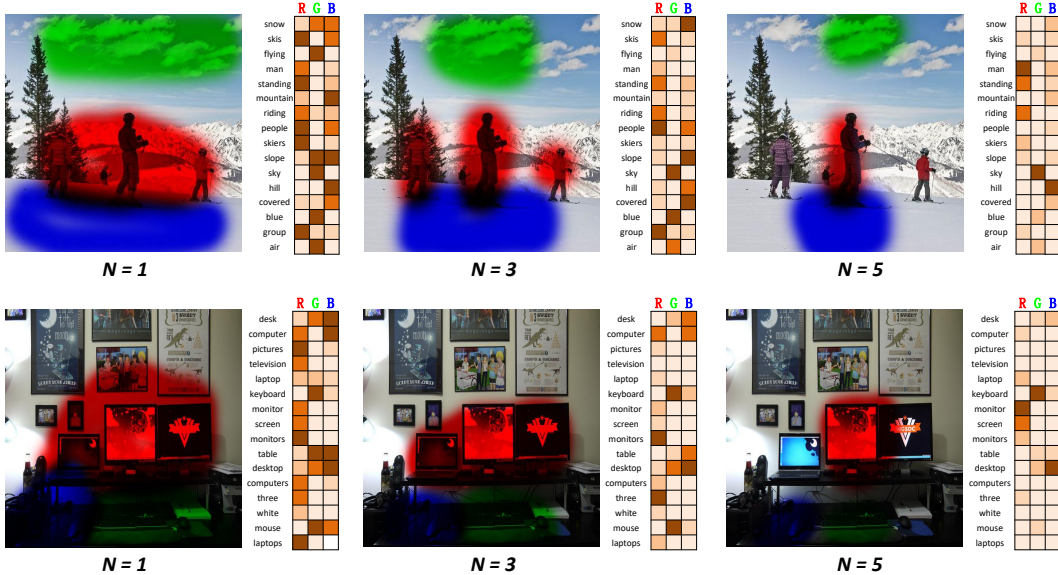

Figure 5: Visualization of the integrated image representations. Please view in color. We show the representations with different iteration $N$ for two images. We choose three visual features and corresponding textual concepts with clear semantic implication and highlight them with distinct colors. As we see, with $N$ increasing, the alignment becomes more focused and more specific, but the combination of related features are less represented.

Table 4, when using the same single-modal features as the corresponding baselines, our method can still promote the performance. Thanks to the mutual iterative attention process, "Visual Attention w/ $I_N$" and "Concept Attention w/ $I_N$" can pay attention to integrated visual features and textual concepts, respectively. This frees the decoder from associating the unrelated original features in each domain, which may explain for the improvements. The performance in terms of SPICE and CIDEr is further elevated when $T_N$ and $I_N$ are combined. The progressively increased scores demonstrate that the improvements indeed come from the refined semantic-grounded image representations produced by MIA, rather than the introduction of additional information.

The SPICE sub-category results show that $I_N$ helps the baselines to generate captions that are more detailed in count and size, $T_N$ results in more comprehensiveness in objects, and MIA can help the baselines to achieve a caption that is detailed in all sub-categories. Due to limited space, the scores are provided in the supplementary materials. For output samples and intuitive comparisons, please refer to the supplementary materials.

**Effect of iteration times.** We select two representative models, i.e., Visual Attention and Concept Attention, to analyze the effect of iteration times. Figure 4 presents the performance of Visual Attention (VA) and Concept Attention (CA) under different evaluation metrics when equipped with the MIA. We evaluate with iteration times ranging from 1 to 5. The scores first rise and then decline with the increase of $N$, as a holistic trend. With one accord, the performances consistently reach the best at the second iteration, for the reason of which we set $N = 2$. It suggests that a single iteration does not suffice to align visual features and textual concepts. With each round of mutual attention, the image representations become increasingly focused, which explains the promotion in the first few iterations. As for the falling back phenomenon, we speculate that the integration effect of MIA can also unexpectedly eliminate some useful information by assigning them low attention weights. The absent of these key elements results in less comprehensive captions. The visualization in Figure 5 also attests to our arguments.

**Visualization.** We visualize the integration of the image representations in Figure 5. The colors in the images and the heatmaps reflect the accumulated attention weights assigned to the original image representations until the current iteration. As we can see in the left plots of Figure 5, the attended visual regions are general in the first iteration, thereby assigning comparable weights to a number of visual words with low relevance. Taking the indoor image as an example, the red-colored visual

region in the left plot focuses not only on the related words (e.g. *computer* and *monitor*) but also the words that describe peripheral objects (e.g. *pictures* on the wall), and words that are incorrect (e.g. *television*). In this case, the inter-domain alignment is weak and the integration of features within a certain domain is not concentrated, making the image representations undesirable. As the two modalities iteratively attend to each other, the features in the two domains gradually concentrate on concrete objects and corresponding visual words. In the third iteration where the model performance peaks (among the visualized iterations), the boundaries of the visual regions are well-defined and the dominant visual words making up the textual concepts are satisfactory. However, the features are over-concentrated in the fifth iteration, filtering out some requisite information. For example, the red region shrinks to a single person in the first example, and a single monitor in the second example, which reduces the information about number (e.g., *group*, *three*, *computers* and *monitors*) and attribute (e.g., *skis*). Hence, it is necessary to decide an appropriate number of iteration for acquiring better image representations.

## 4 Related Work

**Representing images.** A number of neural approaches have been proposed to obtain image representations in various forms. An intuitive method is to extract visual features using a CNN or a RCNN. The former splits an image into a uniform grid of visual regions (Figure 1 (a)), and the latter produces object-level visual features based on bounding boxes (Figure 1 (b)), which has proven to be more effective. For image captioning, Fang et al. [8], Wu et al. [31] and You et al. [35] augmented the information source with textual concepts that are given by a predictor, which is trained to find the most frequent words in the captions. A most recent advance [34] built graphs over the RCNN-detected visual regions, whose relationships are modeled as directed edges in a scene-graph, which is further encoded via a Graph Convolutional Network (GCN).

**Visual-semantic alignment.** To acquire integrated image representations, we introduce the Mutual Iterative Attention (MIA) strategy, which is based on the self-attention mechanism [28], to align the visual features and textual concepts. It is worth noticing that for image captioning, Karpathy and Li [11] also introduced the notion of visual-semantic alignment. They endowed the RCNN-based visual features with semantic information by minimizing their distance in a multimodal embedding space with corresponding segments of the ground-truth caption, which is quite different from our concept-based iterative alignment. In the field of VQA, some recent efforts [19, 12, 6, 20] have also been dedicated to study the image-question alignment. Such alignment intends to explore the latent relation between important question words and image regions. Differently, we focus on a more general purpose of building semantic-grounded image representations through the alignment between visual regions and corresponding textual concepts. The learned semantic-grounded image representations, as shown by our experiments, are complementary to the VQA models that are based on image-question alignment.

## 5 Conclusions

We focus on building integrated image representations to describe salient image regions from both visual and semantic perspective to address the lack of structural relationship among individual features. The proposed Mutual Iterative Attention (MIA) strategy aligns the visual regions and textual concepts by conducting mutual attention over the two modalities in an iterative way. The refined image representations may provide a better start point for vision-and-language grounding problems. In our empirical studies on the MSCOCO image captioning dataset and the VQA v2.0 dataset, the proposed MIA exhibits compelling effectiveness in boosting the baseline systems. The results and relevant analysis demonstrate that the semantic-grounded image representations are essential to the improvements and generalize well to a wide range of existing systems for vision-and-language grounding tasks.

**Acknowledgments**

This work was supported in part by National Natural Science Foundation of China (No. 61673028). We thank all the anonymous reviewers for their constructive comments and suggestions. Xu Sun is the corresponding author of this paper.

## Footnotes

[3]We build the shortcut connection by adding the source features to the sub-layer outputs, instead of the querying features in Vaswani et al. [28], to ensure no heterogeneous information is injected.

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
