[Supplementary Material · NeurIPS_2019_Supp.pdf]

# Aligning Visual Regions and Textual Concepts for Semantic-Grounded Image Representations
# - Supplementary Materials

**Fenglin Liu[1]**∗ **Yuanxin Liu[3,4]**∗ **Xuancheng Ren[2]**∗ **Xiaodong He[5], Xu Sun[2]**
[1]ADSPLAB, School of ECE, Peking University, Shenzhen, China
[2]MOE Key Laboratory of Computational Linguistics, School of EECS, Peking University
[3]Institute of Information Engineering, Chinese Academy of Sciences
[4]School of Cyber Security, University of Chinese Academy of Sciences
[5]JD AI Research
{fenglinliu98, renxc, xusun}@pku.edu.cn, liuyuanxin@iie.ac.cn
xiaodong.he@jd.com

## A  Experiments on Image Captioning

### A.1  Dataset and Evaluation Metrics

MSCOCO [5] is a popular dataset for image captioning. It contains 123,287 images, each of which is paired with 5 descriptive sentences. Following common practice [16, 2, 15, 14], we report results with the help of the MSCOCO captioning evaluation toolkit [5], and use the publicly-available splits provided by Karpathy and Li [8], where the validation set and test set both contain 5,000 images. The toolkit includes the commonly-used evaluation metrics SPICE, CIDEr, BLEU, METEOR and ROUGE in image captioning task. SPICE [1] and CIDEr [19] are customized for evaluating image captioning systems, based on scene-graph matching and n-gram matching, respectively. BLEU [17] and METEOR [4] are originally designed for machine translation, and ROUGE [11, 12] measures the quality of summaries.

### A.2  Baselines and Implementation Details

We design five representative baselines, which are built on the models in previous work. Especially, since our main contribution is to provide a new kind of image representations, those baselines use different kinds of image representations, and we keep the backbone of those models as neat as possible.

The baselines are described as follows:

• **Visual Attention.** The Visual Attention model is adapted from the spatial attention model in Lu et al. [16]. It uses the 49 grid visual features $I \in \mathbb{R}^{L_I \times d_h}$ from the last convolutional layer of ResNet-152 as the image representation. The decoder is a LSTM model initialized with two zero vectors: $h_0 = \mathbf{0}, c_0 = \mathbf{0}$, which is the same for the other baselines, with the exception that Visual Regional Attention contains two LSTM decoders. For each decoding step, the decoder takes the caption embedding $w_t^e$, added with the averaged visual features $I_a = \frac{1}{L_I} \sum_{i=1}^{L_I} I_i$ as input to the LSTM:

$$h_t = \text{LSTM}\left(h_{t-1}, w_t^e + I_a\right) \tag{1}$$

Then, the LSTM output $h_t$ is used as a query to attend to the visual features:

$$\alpha_t = \text{softmax}\left(w_\alpha \tanh\left(W_I I^{\text{T}} \oplus W_h h_t\right)\right), \quad c_t = \alpha_t I \tag{2}$$

---

∗Equal contribution.

where the $w_\alpha$, $W_I$ and $W_h$ are the learnable parameters. $\oplus$ denotes the matrix-vector addition, which is calculated by adding the vector to each column of the matrix. Finally, the LSTM output and the attended visual features are used to predict the next word:

$$y_t \sim p_t = \text{softmax}\left(W_p\left(h_t + c_t\right)\right) \tag{3}$$

To augment Visual Attention with MIA, both the visual features in Eq. (1) and Eq. (2) are replaced with the MIA-refined image representations. For "Visual Attention w/ $I_N$", we only use MIA-refined visual features $I_N$ for the same replacement.

• **Concept Attention.** The Concept Attention model is built by adapting the semantic attention model in You et al. [22]. We predict 49 textual concepts $T \in \mathbb{R}^{L_T \times d_h}$ using a textual concept extractor proposed by Fang et al. [6]. Similar to You et al. [22], the averaged visual features $I_a$ is fed to the decoder at the first time step, which means $h_1 = \text{LSTM}(h_0, I_a)$. For the subsequent decoding steps, the input is a sum of the caption embedding $w_t^e$ and the averaged textual concepts $T_a = \frac{1}{L_T}\sum_{i=1}^{L_T} T_i$:

$$h_t = \text{LSTM}\left(h_{t-1}, w_t^e + T_a\right) \tag{4}$$

An attention operation is performed over the textual concepts, with $h_t$ as the query:

$$\alpha_t = \text{softmax}\left(w_\alpha \tanh\left(W_T T^{\text{T}} \oplus W_h h_t\right)\right), \quad c_t = \alpha_t T \tag{5}$$

Then, like Eq. (3), a softmax layer predicts the output word distribution. For the deployment of MIA, the original textual concepts are replaced with refined image representations or refined textual concepts $T_N$, in similar fashion as for the original visual features.

• **Visual Condition.** The Visual Condition model is adapted from the LSTM-A4 model in Yao et al. [21], jointly considers the visual features and textual concepts. Specifically, Visual Condition refers to the averaged textual concepts $T_a$ at the first decoding step, while takes the sum of the averaged visual features $I_a$ and the word embedding $w_t^e$ as input for the subsequent steps:

$$h_1 = \text{LSTM}\left(h_0, T_a\right) \tag{6}$$
$$h_t = \text{LSTM}\left(h_{t-1}, w_t^e + I_a\right), \quad t \geq 2 \tag{7}$$

The LSTM output $h_t$ is then followed with a softmax layer to predcit the next word:

$$y_t \sim p_t = \text{softmax}\left(W_p h_t\right) \tag{8}$$

After being processed by MIA, the refined image representations are used as replacement for both the visual features and textual concepts in Eq. (6) and Eq. (7), respectively.

• **Concept Condition.** The Concept Condition model is adapted from the LSTM-A5 model in Yao et al. [21]. In contrast to Visual Condition, it reverses the order of input visual features and textual concepts, which can be defined as follows:

$$h_1 = \text{LSTM}\left(h_0, I_a\right) \tag{9}$$
$$h_t = \text{LSTM}\left(h_{t-1}, w_t^e + T_a\right), \quad t \geq 2 \tag{10}$$

The prediction of the next caption word and the utilization of MIA are also similar to Visual Condition.

• **Visual Regional Attention.** The Visual Regional Attention model is adapted from the Up-Down model in Anderson et al. [2]. It uses the 36 region-based visual features, which are extracted by a variant of Faster R-CNN [18]. The Faster R-CNN is provided by Anderson et al. [2] and is pre-trained on Visual Genome [10]. Two stacked LSTMs are adopted for caption generation, both of which are initialized with zero hidden states: $h_0^1 = \mathbf{0}, c_0^1 = \mathbf{0}; h_0^2 = \mathbf{0}, c_0^2 = \mathbf{0}$. At each decoding step, the first LSTM takes the caption embedding $w_t^e$, concatenated with the averaged visual features $I_a$ as input:

$$h_t^1 = \text{LSTM}_1\left(h_{t-1}^1, [w_t^e; I_a]\right) \tag{11}$$

Then, the first LSTM output $h_t^1$ is used as a query to attend to the region-based visual features:

$$\alpha_t = \text{softmax}\left(w_\alpha \tanh\left(W_I I^{\text{T}} \oplus W_h h_t^1\right)\right), \quad c_t = \alpha_t I \tag{12}$$

After that, the second LSTM takes the first LSTM output $h_t^1$, concatenated with the attended visual features $c_t$ as input, followed by a softmax layer to predict the target word:

$$h_t^2 = \text{LSTM}_2\left(h_{t-1}^2, [h_t^1; c_t]\right) \tag{13}$$

$$y_t \sim p_t = \text{softmax}\left(W_p h_t^2\right) \tag{14}$$

For the MIA-augmented model, we replace the region-based visual features in Eq. (11) and Eq. (12) with MIA-refined image representations.

# B Experiments on Visual Question Answering

## B.1 Dataset and Evaluation Metrics

We evaluate the models on VQA version 2.0 [7], which is comprised of image-based question-answer pairs labeled by human annotators, where the images are collected from the MSCOCO dataset [13].

VQA v2.0 is an updated version of previous VQA 1.0 with much more annotations and less dataset bias. VQA v2.0 is split into train, validation and test sets. There are 82,783, 40,504 and 81,434 images, (443,757, 214,354 and 447,793 corresponding questions) in the training, validation and test set, respectively. The questions are categorized into three types, namely *Yes/No*, *Number* and *other* categories. Each question is accompanied with 10 answers composed by the annotators. Answers with the highest frequency are treated as the ground-truth. Evaluation is conducted on the test set, the reported accuracies are calculated by the standard VQA metric [3], with occasional disagreement between annotators being considered.

## B.2 Baselines and Implementation Details

We choose Up-Down [2] and BAN [9] for comparison, where the former is the winner of VQA challenge 2017 and the latter is the state-of-the-art on VQA v2.0 dataset. They both use region-based visual features as image representations and GRU-encoded hidden states as question representations, and make classification based on their combination. However, Up-Down only uses the final sentence vector to obtain the weight of each visual region, while BAN uses a bilinear attention to obtain the weight for each pair of visual region and question word. For equipping with our MIA, we simply replace the original visual features with semantic-grounded image representations provided by MIA.

# C Further Experimental Analysis

## C.1 Effect of Guiding Scheme

We can either start with the textual concepts guiding the integration of the visual features or let the latter to take the initiative. Even if the role of visual features and textual concepts are equivalent in mutual attention, the choice of guiding scheme could make a difference. We examine the performance of Visual Attention and Concept Attention when the visual features first attend to the textual concepts. As shown in Table 1, the model scores are inferior to that of the alternative scheme. Especially, the performance of the Visual Attention has even been impaired. The rationale for such phenomenon is presumably the limited visual receptive field of the original visual features, which makes them inadequate to integrate the textual concepts. As to the textual concepts, they are inherently good at describing integrated visual regions, as they contain high-level semantic information.

Table 1: Evaluation of different guiding scheme.

| Model | BLEU-1 | BLEU-2 | BLEU-3 | BLEU-4 | METEOR | ROUGE | CIDEr | SPICE |
|---|---|---|---|---|---|---|---|---|
| Visual Attention | 72.6 | 56.0 | 42.2 | 31.7 | 26.5 | 54.6 | 103.0 | 19.3 |
| $I_N$ -> $T_N$ | 73.2 | 56.8 | 42.8 | 32.0 | 25.5 | 53.9 | 99.0 | 18.7 |
| $T_N$ -> $I_N$ | **74.5** | **58.4** | **44.4** | **33.6** | **26.8** | **55.8** | **106.7** | **20.1** |
| Concept Attention | 72.6 | 55.9 | 42.5 | 32.5 | 26.5 | 54.4 | 103.2 | 19.4 |
| $I_N$ -> $T_N$ | 73.2 | 56.5 | 43.0 | 32.9 | 26.6 | 54.7 | 105.5 | 19.5 |
| $T_N$ -> $I_N$ | **73.8** | **57.4** | **43.8** | **33.6** | **27.1** | **55.3** | **107.9** | **20.3** |

## C.2   Effect of Incorporating MIA with Scene-Graph based Models

SGAE$_{fuse}$ [20], which learns finer representations of an image through scene-graphs, is the state-of-the-art image captioning system at the time of our submission. We incorporate the scene-graph based features with MIA-refined image representations, and see whether MIA can still help SGAE. As presented in Table 2, MIA also boosts the performance of SGAE, indicating that MIA learns very effective representations even for scene-graphs.

Table 2: Evaluation of on the scene-graph based model.

| Methods | BLEU-4 | METEOR | ROUGE | CIDEr | SPICE |
|---|---|---|---|---|---|
| SGAE$_{fuse}$ | 39.3 | 28.5 | 58.8 | 129.6 | 22.3 |
| w/ MIA | **39.6** | **29.0** | **58.9** | **130.1** | **22.8** |

## C.3   SPICE Sub-Category Results

For a better understanding of the differences of the generated captions by different methods, we report the breakdown of SPICE F-scores (see Table 3). As we can see, the $I_N$, $T_N$ and MIA promotes the baselines over almost all sub-categories. Especially, the $I_N$ is good at associating related parts in the image, which is demonstrated by the increased scores in *Count* and *Size*. and the $T_N$ collects relevant textual concepts, providing comprehensive context that is detailed in objects. Encouragingly, when incorporating $I_N$ and $T_N$ at the same time, i.e., *w/ MIA*, the advantages of the $I_N$ and $T_N$ are united to produce a balanced improvement. It proves the effectiveness of our approach.

Table 3: Variation of model performance under the breakdown of SPICE F-scores. We can find that the w/ $T_N$ has a higher *Object* scores than the baselines, and the w/ $I_N$ reaches better scores in *Count* and *Size*. As we can see, incorporating Mutual Iterative Attention (MIA) directly on the baselines, leads to overall improvements.

| Methods | SPICE | | | | | | |
|---|---|---|---|---|---|---|---|
| | All | Objects | Attributes | Relations | Color | Count | Size |
| Visual Attention | 19.3 | 35.2 | 9.1 | 5.3 | 10.7 | 3.0 | 3.3 |
| w/ $I_N$ | 19.6 | 35.8 | 9.3 | 5.5 | 9.9 | **7.2** | **4.2** |
| w/ MIA | **20.1** | **36.4** | **9.8** | **5.7** | **10.8** | 6.9 | 3.9 |
| Concept Attention | 19.4 | 34.8 | 10.3 | 5.3 | 13.5 | 4.7 | 4.8 |
| w/ $T_N$ | 20.0 | 35.8 | 10.7 | 5.4 | 13.6 | 4.2 | 4.6 |
| w/ MIA | **20.3** | **36.1** | **11.4** | **5.5** | **14.1** | **7.1** | **5.2** |

## C.4 Samples of Generated Captions

We show the captions generated by the method *w/o MIA* and the method *w/ MIA* to intuitively analyze the differences of the methods. As shown in Table 4, the *w/ $I_N$* is good at portraying the number and size but is less specific in objects. The *w/ $T_N$* includes more objects but lacks details, such as number. The proposed MIA can help the baselines to achieves a very good balance.

Table 4: Examples of the captions generated by different methods. For every example, we show the top-10 relevant textual concepts. Based on the Mutual Iterative Attention (MIA) over the source information, from the generated captions, we can find that the *w/ $T_N$* results in more comprehensiveness in objects. The *w/ $I_N$* helps the baselines to generate captions that are more detailed in count and size, and the *w/ MIA* is able to generate more complete captions that is detailed both in the objects, attributes, relations and color.

| Image | Concepts | Captions |
|---|---|---|
| *Visual Attention (Based on Visual Features)* | | |
|  | water boat luggage sitting black ocean large white suitcases near | Reference: a number of suitcases on the boat in the sea. |
| | | Baseline: a suitcase sitting on top of a body of water. |
| | | w/ $I_N$: a couple of luggage sitting on top of a boat. |
| | | w/ MIA: a couple of black luggage sitting on the edge of the water. |
|  | standing zebras zebra field grass dry tall close stand large | Reference: two zebras stand in a field with tall grass. |
| | | Baseline: a zebra standing in the middle of a field. |
| | | w/ $I_N$: two large zebras standing in a grass field. |
| | | w/ MIA: a couple of zebras standing on top of a dry grass field. |
| *Concept Attention (Based on Textual Concepts)* | | |
|  | vase flowers table glass display sitting orange filled red yellow | Reference: orange, red and white flowers in vases on tables. |
| | | Baseline: a vase filled with some orange flowers. |
| | | w/ $T_N$: a vase filled with yellow flowers on top of a table. |
| | | w/ MIA: a small vase filled with red and orange flowers on a table. |
|  | bus double decker red street down city road driving stop | Reference: a red double decker bus is driving on a city street. |
| | | Baseline: a red bus driving down a street. |
| | | w/ $T_N$: a double decker bus driving down a city street. |
| | | w/ MIA: a red double decker bus driving down a city street. |