[Reviews · NeurIPS 2019]

Reviewer 1



This paper describes a method for integrating visual and textual features within a self-attention-like architecture. Overall I find this to be a good paper presenting an interesting method, with comprehensive experiments demonstrating the capacity of the method to improve on a wide range of models in image captioning as well as VQA.The analysis is informative, and the supplementary materials add further comprehensiveness. My main complaint is that the paper could be clearer about the current state of the art in these tasks and how the paper's contribution relates to that state of the art. The paper apparently presents a new state-of-the-art on the COCO image captioning dataset, by integrating the proposed method with the Transformer model. It doesn't, however, report what happens if the method is integrated with the prior state-of-the-art model SGAE -- was this tried and shown not to yield improvement? I also found it odd that mention of the identity of the current state-of-the-art system, and the authors' surpassing of that state-of-the-art, was limited to a caption of one of the tables and not included in the text of the Experiments section (though the new state-of-the-art is mentioned in the intro and conclusion). Making all of this more explicit in the text of the Experiments section would be helpful for contextualizing the results. Also, because most of the reported results are showing improvements over systems that are not state-of-the-art, it would be good to be clear about the importance of showing these results and what we should take away from them. Are these still strong baselines? Is it simply useful to show that the refined representations are useful across these different categories of systems? Similarly, though the VQA section does identify a state-of-the-art model in the text and report results surpassing that model, the results are on the validation set, so we are left without knowing how this method contributes to performance on test relative to the state-of-the-art. The supplementary materials mention that this has to do with difficulties in running on the test set, but this is a bit unsatisfying. Minor: line 188 overpasses --> overtakes/surpasses line 207: that most --> that are most line 236: tailored made --> tailored (or tailor-made) line 242: absent --> absence

Reviewer 2



The overall idea is interesting and experiments show that it constantly improves the performance of two major integrated tasks, image captioning and visual question answering. The presentation of MIA is clearly described, and it is understandable to use the method for pretraining of integrated features. However, when it is applied to image captioning where the inputs are only images, it is a bit difficult how those features are used. It is because Figure 3 and its explanation are too brief. It is preferable the architecture of LSTM-A3 is explained or illustrated in some detail.

Reviewer 3



-- What is the neural structure of the component that processes textual concepts? Bag-of-words, bag-of-embeddings or RNN encoder without order? -- How is the textual concept extractor carried out? Is it basically the extractor proposed by Fang et al. (2015) or Wu et al. (2016)? What are the datasets used for training the textual concepts? --Line 107. Typo, "... from the self-attention in that in the self-attention, the query matrix" -> "... from the self-attention where the query matrix" Citations: Lu, Jiasen, et al. "Hierarchical question-image co-attention for visual question answering." Advances In Neural Information Processing Systems. 2016. Xiong, Caiming, Victor Zhong, and Richard Socher. "Dynamic coattention networks for question answering." arXiv preprint arXiv:1611.01604 (2016).

[Author Response · NeurIPS 2019]

Table 1: Evaluation of the state-of-the-art model SGAE on the image captioning dataset.

| Model | B@4 | M | R | C | S |
|---|---|---|---|---|---|
| SGAE | 38.8 | 28.5 | 58.7 | 129.3 | 22.3 |
| w/ MIA | **39.6** | **29.0** | **58.9** | **130.1** | **22.8** |

Table 2: The accuracy on the VQA v2.0 test set.

| Model | Number | Other | Overall | Test-std |
|---|---|---|---|---|
| Up-Down | 44.2 | 56.2 | 65.6 | 65.9 |
| w/ MIA | **51.2** | **59.7** | **68.8** | **69.1** |
| BAN | 50.9 | 60.3 | 69.6 | 69.8 |
| w/ MIA | **53.1** | **60.5** | **70.2** | **70.3** |

We thank all the reviewers for the helpful comments. As we replied, we will revise the paper to address the concerns and further proofread the paper.

**Q1: How the paper's contribution relates to the current SOTA?**

**A1:** For image captioning, we simply had not tried to combine MIA with SGAE [2], because our main goal is to show that MIA could improve the performance of a wide range of established models. SGAE is a rather complicated scene-graph based method specific to image captioning. But it is indeed interesting to see whether MIA can still help SGAE since scene-graphs are also finer representations of an image. As presented in Table 1, it is clear that MIA also boosts the performance of SGAE (a higher new SOTA), indicating that MIA learns very effective representations even for scene-graphs.

For VQA, as shown in Table 2, the two state-of-the-art systems for VQA also achieve an overall improvement on the test set, which is in accordance with the results on the validation set in the paper.

The results with current SOTA + MIA will be stated more clearly in the paper.

**Q2: How to use MIA on the baseline systems (i.e., how is MIA applied to image captioning where the inputs are only images) and what are the experimental settings?**

**A2:** We can extract visual and textual features (see Figure 1 in the paper) for an image, regardless of the tasks (image captioning or VQA). This indicates that MIA does not require original text information about the image (e.g., the questions in VQA), which is not provided in the task of image captioning. For using the semantic-grounded features learned by MIA, we can simply replace the original source features with MIA-refined features. Specifically, in Figure 3 of the paper, traditional LSTM-A3 takes as input textual features at the first decoding step and visual features at the second step. Since MIA will not change the number or the size of the feature vectors (each of them can be seen as a weighted average of the original features), we can replace the original features directly.

For the settings, we have listed them in the supplementary materials. In particular, we preserve the original settings for all the baseline models, since our focus is to provide better image representations.

**Q3: How is the textual concept extracted and processed?**

**A3:** We predict textual concepts using the textual concept extractor proposed by Fang et al. (2015) [1], which is based on fully convolutional network (FCN) and is purely trained on the training data of COCO image captioning dataset for concept prediction, i.e., we do not adopt the ground-truth textual concepts (if the ground-truth textual concepts are utilized, the SGAE w/ MIA model can achieve a CIDEr score of 168.7, which indicates that better textual concepts will give rise to better performance of downstream tasks). As shown in line 83 of the paper, we simply apply word embedding (randomly initialized), which is shared with the caption/question words, to process textual concepts. It is worth noticing that in our baselines, no further processing except MIA is done. Besides, we do not use extra dataset to train MIA. The MIA is trained jointly with the baseline models from scratch, therefore no external knowledge is introduced to boost the performance. Especially, as presented in Table 3 of the paper, when applied to LSTM-A3 and LSTM-A4, which use the same (visual and textual) features as we adopted, MIA can still boost the performance, further demonstrating the effectiveness of alignment between visual regions and textual concepts.

# References

[1] H. Fang, S. Gupta, F. N. Iandola, R. K. Srivastava, L. Deng, P. Dollár, J. Gao, X. He, M. Mitchell, J. C. Platt, C. L. Zitnick, and G. Zweig. From captions to visual concepts and back. In *CVPR*, 2015.

[2] X. Yang, K. Tang, H. Zhang, and J. Cai. Auto-encoding scene graphs for image captioning. In *CVPR*, 2019.



[Meta-Review · NeurIPS 2019]

The paper proposes a new method called Mutual Iterative Attention (MIA) for improving the representations used by common visual-question-answering and image captioning models. MIA works by repeated execution of 'mutual attention', a computation that is similar to the self-attention operation in the Transformer model, but where the lookup ('query') representation is conditioned by information from the other modality. Importantly, the two modalities involved in the MIA operation are not vision and language, they are vision and 'textual concepts' (which they also call 'textual words' and 'visual words' at various points in the paper). These are actual words referring to objects that can be found in the image. The model that predicts textual concepts (the 'visual words' extractor) is trained on the MS-COCO dataset in a separate optimization to the captioning model Applying MIA to a range of models before attempting VQA or captioning tasks improves the scores, in some cases above the state-of-the-art. It is a strength of this paper that the authors apply their method to a wide range of existing models and observe consistent improvements. This indicates the promise of the work, and it seems to me that for this reason the reviewers have recommended that the paper be accepted. However, the reviewers have also raised some concerns, which I share. As with each of the other reviewers, I find that the overall method is not clearly explained in the paper. It took me many readings to reach some understanding of what textual concepts were (partly because the authors give them different names in different parts of the paper). The authors apply MIA to many different models, each of which works in quite a different way, and it is (still) not clear to me exactly how it interacts with each of these existing models. As a concrete example, despite helpful discussions with the Reviewers, we are all still somewhat confused by Tables 1-3. Is it really possible to train a captioning model without access to any visual features (i.e. based solely on 'textual concepts')? This may be the case, but if so this must be much more clearly explained to a reader not familiar with the application of textual concepts to captioning (without visual features). Another concern is that textual concepts typically requires more training data (the data needed to train the extractor). As the authors point out in their rebuttal, for MS-COCO task, this is not the case, since the extractor was trained on precisely the MS-COCO training data. However, for the other tasks, it is fair to say that more data is being used to train a model that uses textual concepts. I think this is in fact an interesting application of transfer learning, but, as mentioned above, this is not made clear in the paper. It took me a long time to work out that this is what was going on, and in my opinion this *must* be discussed more explicitly and openly in the paper for the work to meet the standards of transparency and clarity expected of Neurips. In short, I agree with the reviewers that this is a promising method that can improve image captioning and VQA systems (and potentially any models that rely on mixing vision and language). However, if the reviewers recommendations are followed and the paper is accepted then it is my recommendation that the authors must comprehensively re-write parts of the paper to give a clear explanation of a) what textual concepts are, b) how a captioning model can be trained directly from them (without access to the underlying image) c) how much data is required to train a textual concept extractor and d) how exactly MIA is applied with the range of existing models considered in the paper.